# A_2A_ Adenosine Receptor as a Potential Biomarker and a Possible Therapeutic Target in Alzheimer’s Disease

**DOI:** 10.3390/cells10092344

**Published:** 2021-09-07

**Authors:** Stefania Gessi, Tino Emanuele Poloni, Giulia Negro, Katia Varani, Silvia Pasquini, Fabrizio Vincenzi, Pier Andrea Borea, Stefania Merighi

**Affiliations:** 1Department of Translational Medicine, University of Ferrara, 44121 Ferrara, Italy; vrk@unife.it (K.V.); psqslv@unife.it (S.P.); vncfrz@unife.it (F.V.); bpa@unife.it (P.A.B.); 2Department of Neurology & Neuropathology, Golgi-Cenci Foundation, 20081 Abbiategrasso, Italy; e.poloni@golgicenci.it (T.E.P.); g.negro4@campus.unimib.it (G.N.); 3Department of Neurology, School of Medicine and Surgery, University of Milano-Bicocca, 20126 Monza, Italy

**Keywords:** Alzheimer’s disease, biomarkers, adenosine A_2A_ receptor

## Abstract

Alzheimer’s disease (AD) is one of the most common neurodegenerative pathologies. Its incidence is in dramatic growth in Western societies and there is a need of both biomarkers to support the clinical diagnosis and drugs for the treatment of AD. The diagnostic criteria of AD are based on clinical data. However, it is necessary to develop biomarkers considering the neuropathology of AD. The A_2A_ receptor, a G-protein coupled member of the P1 family of adenosine receptors, has different functions crucial for neurodegeneration. Its activation in the hippocampal region regulates synaptic plasticity and in particular glutamate release, NMDA receptor activation and calcium influx. Additionally, it exerts effects in neuroinflammation, regulating the secretion of pro-inflammatory cytokines. In AD patients, its expression is increased in the hippocampus/entorhinal cortex more than in the frontal cortex, a phenomenon not observed in age-matched control brains, indicating an association with AD pathology. It is upregulated in peripheral blood cells of patients affected by AD, thus reflecting its increase at central neuronal level. This review offers an overview on the main AD biomarkers and the potential role of A_2A_ adenosine receptor as a new marker and therapeutic target.

## 1. Alzheimer’s Disease 

According to DSM-V, the mental deficit described as neurocognitive disorder (NCD) is caused by pathologies affecting neuronal circuits. Then, the type of underlying pathology defines the etiology of the disorder. While the early stages of NCD (mild-NCD/MCI) are characterized by functional preservation of everyday activities, major-NCD (dementia) has a functional impact on daily life [1].

The main age-related NCD is Alzheimer’s disease (AD); AD pathology progressively involves all cortical areas, and during its evolution, all cognitive domains are weakened with profound changes in behavior and functional abilities. Research over the last few decades has led to the discovery of some risk factors, including non-modifiable genetic factors (AD-related polymorphisms, APO-E4 allele, and pathogenic mutations in PSN-1-2 and APP genes) [2,3] and modifiable factors (favorable behaviors: regular physical and mental activity, healthy diet, high education and social engagement, and harmful conditions: midlife obesity, hypertension, diabetes, smoke, excessive alcohol and hearing loss) [4,5]. Nonetheless, the pathogenetic trajectory of AD is complex and individual, and largely unknown; hence, there is a need to assess the neuropathological picture and the related biomarkers to explore etiopathogenesis.

According to the amyloidogenic theory, the extensive presence of cortical amyloid and, therefore, of toxic β-amyloid oligomers induces a synaptic and neuronal dysfunction, mainly through TAU protein hyperphosphorylation, in turn causing synaptic collapse, fibers degeneration and neuronal loss. Furthermore, amyloid species seem to determine a reduction in the blood flow of brain capillaries and glial inflammatory activation [6,7]. Together, these neurodegenerative processes determine both macroscopic and microscopic brain changes. Typically, progressive brain atrophy is observed, starting in the parahippocampal cortex, hippocampus, medial and basal temporal lobe and parietal lobe, and spreading to the whole cortex in the advanced stages of the disease [8,9]. The underlying microscopic features are characterized by a dual proteinopathy, which consists of the deposition of amyloid and phosphorylated TAU (pTAU). The first proteinopathy is characterized by cortical plaques (β-amyloid or senile plaques) composed by an extracellular accumulation of β-amyloid peptides; the second one is represented by the hyperphosphorylated tau protein that aggregates inside the dying neuron, generating the so-called neurofibrillary tangles (NFT) and neuropil threads (NTs). The combination of the two proteinopathies constitutes the neuritic plaque (NP), which is the hallmark of AD neuropathology. Therefore, the aggregate scores for Amyloid (Thal stages), TAU (Braak stages) and NP (CERAD grading) constitute the ABC criteria for AD pathology, and define the neuropathological diagnosis of AD [10,11,12,13].

## 2. Biomarkers of AD

As stated previously, AD is defined as a clinical-pathologic entity, which is diagnosed during the course of life as possible or probable disease, and definitively at autopsy [14]. In order to comprehend the mechanism underlying the clinical expression of AD, a biological- as well as a syndrome-based definition is necessary. Furthermore, in order to identify therapies or interventions that prevent or delay the initial onset of symptoms, a biological-based definition including the preclinical phase of the disease is pivotal. Several studies are attempting to find novel biomarkers that reflect the biology of the disease, improving current diagnosis across the AD continuum [15]. 

Currently, various CSF and imaging biomarkers reflecting neuropathological changes are widely used in AD. An unbiased descriptive classification scheme for the biomarkers named “ATN” system was proposed; it includes seven major AD biomarkers divided in three categories based on the nature of the pathologic process measured by them. Biomarkers of β-amyloid plaques, labeled as “A”, are related to the extent of cortical amyloid deposition, and demonstrated by amyloid-PET or low CSF values of β-amyloid 42 (increased brain amyloid production and/or reduced amyloid “cleaning”). Biomarkers of pTAU, labeled as “T”, are related to cortical pTAU deposition and determined through TAU-PET or elevated CSF pTAU. Biomarkers of neurodegeneration and neuronal injury, labeled as “N”, are expressed by the demonstration of decreased synaptic activity through [(18)F]-fluorodeoxyglucose PET (FDG-PET hypometabolism), volume reduction in specific regions of interest (ROI) of the brain (atrophy on CT/MRI), or total-TAU release from dying neurons (increase in CSF total-TAU, as an effect of neuronal lysis). Each biomarker is evaluated positive or negative and, throughout their combination, the individual status of the biomarkers is defined; in this way, three categories are identifiable: (1) individuals with normal AD biomarkers; (2) those in the Alzheimer’s continuum, characterized by the presence of a significant amyloid load, and divided in amyloid deposition only, amyloid deposition and non-TAU neurodegeneration (suspected non-AD pathology), amyloid deposition and TAU pathology (early AD and AD); (3) non-Alzheimer’s neurodegenerative diseases, including those with no amyloid deposition (normal “A” biomarkers) but showing TAU pathology and/or neurodegeneration (abnormal “T” and/or “N”) [15,16], see Table 1. 

The last category has been recently identified thanks to the use of biomarkers; it includes some forms of dementia of very elderly people with amnesic pictures, of varying severity, which mimic AD but have a different biological history. These pathologies mainly involve the temporo-mesial structures and are associated with the deposition of pTAU in the absence of amyloid, as occurs in primary TAUopathies such as primary age-related TAUopathy (PART) or argyrophilic grain disease [17], or with the presence of non-TAU neurodegeneration, as occurs in limbic-predominant age-related TDP-43 encephalopathy (LATE) [18].

The ATN system represents the set of markers most directly linked to the neuropathological picture and, therefore, constitutes the core biomarkers in neurodegeneration. However, the compensatory capacity of the brain and the cognitive reserve should be taken into consideration because protein deposits and atrophic aspects are linked to aging regardless of the presence of a clinically relevant disease, which can be prevented or delayed by the cognitive reserve. Indeed, the ATN system may have a high sensitivity but a low specificity, and, precisely for this reason, it should be interpreted in the light of the clinical and neuropsychological picture. For instance, according to the Cochrane review, amyloid-PET has a sensitivity of 95% and a specificity of 60%, while the FDG-PET has a sensitivity of 75% and a specificity of 85%. The combination of the two exams is more accurate but much more expensive; indeed, both are not recommended for routine diagnostics [19,20]. However, the sensitivity and specificity for AD diagnosis slightly improve using CSF to detect the ATN markers [21]. Therefore, the best diagnostic accuracy is obtained combining different assessments.

The “ATN” system utilizes expensive cerebral imaging or invasive procedures, such as lumbar puncture to obtain CSF. Thus, researchers are looking for AD biomarkers obtainable from more easily accessible biological fluids, such as plasma or serum. The most promising new AD fluid biomarkers include the plasmatic determination of neurofilaments, pTAU species, Aβ-amyloid oligomers (AβOs) and Aβ42/Aβ40 ratio; they might also be used as screening to select cases for further investigation through the ATN system. Overall, the reported sensitivity and specificity of these new fluid markers are similar, both being between 70% and 90% [22,23,24,25,26], but their diagnostic accuracy is still under investigation. The neurofilaments light (NfL) marker consists of light protein chains indicative of axonal damage, whose concentration is increased in the plasma samples of the early stage of AD and increased over time [22,27]. Assays have been developed for the detection of blood pTAU phosphorylated at threonine 181 (pTAU-181), which is increased along the AD continuum and allows the differentiation between AD and non-AD neurodegenerative diseases. Moreover, plasma pTAU-181 is strictly related to its increase in the CSF, and it predicts positive TAU-PET scans [23]. Besides pTAU-181, plasma levels of pTAU phosphorylated at threonine-217 (pTAU-217) is a new candidate tool as biomarker of AD; an increase occurs in the early stage of AD and correlates with worsening of cognition and brain atrophy [28]. Multimeric Detection System (MDS) is a new enzyme-linked immunosorbent assay used to detect AβOs selectively in the plasma of patients with AD. AβOs are the toxic forms of Aβ peptides and their plasma level is higher compared to controls without AD [24]. According to different studies, a significantly lower level of plasmatic Aβ42/Aβ40 ratio (TP42/40) was found in MCI patients compared to healthy controls. Furthermore, TP42/40 inversely correlates with the neocortical amyloid deposition evaluated with amyloid-PET, and was in accordance with the AD biomarkers in CSF [25,26].

Furthermore, Neurogranin (Ng) and inflammatory markers should be mentioned. Ng is a postsynaptic protein, which is increased in the CSF of patients with AD and is supposed to predict the decline in memory and executive function during the early stage of the disease [29]. Regarding inflammation and its important role in AD pathogenesis and its clinical worsening [30,31,32], it should be considered that there are no inflammatory markers in body fluids currently recognized as diagnostic for AD. Reports suggest a possible role of CSF proinflammatory cytokines levels, such as TNF-α, IL-1β or IL-6, as biomarkers of conversion of MCI to AD [33]. However, common inflammatory biomarkers are related to a state of systemic inflammation. Thus, the information obtained from these markers is non-specific and scarcely indicative of what happens in the brain tissue. Therefore, current efforts are moving towards specific markers of microglial/astroglial activation, such as YKL-40 (also known as chitinase-3-like protein), a tracking biomarker of astroglial-related neuroinflammation, which is increased in CSF of AD patients contributing to differentiate AD from non-AD pathologies [34,35], see Table 2.

Body fluid biomarkers play a continuously more important role in clinical trials; they may be used to assess the biological response to therapy and, thus, its efficacy on the course of the disease [36]. The recent approval of Aducanumab by the Food and Drug Administration has brought to the fore the importance of biomarkers to follow AD progression from a biological point of view. Indeed, Aducanumab appears to be more effective on biomarkers than on clinical manifestations, being very effective in removing amyloid and lowering pTAU, but with a modest clinical benefit, which remains uncertain in the long term [37,38,39]. This raises the still-unsolved problem of how much the clinical course of neurodegenerative diseases is predictable through the use of biomarkers. Given the pathogenetic complexity of these diseases, this is not surprising. On the other hand, the study of biomarkers provides indispensable interpretative keys. The study of adenosine receptors fits into this framework; they are very important in the functionality of the hippocampus, which is the nodal center for various neurodegenerative diseases, particularly for AD.

## 3. A_2A_ Adenosine Receptors Biology

A_2A_ receptors belong to the family of G-protein coupled purinergic P1 proteins, including four subtypes, named A_1_, A_2A_, A_2B_ and A_3_, activated by the ubiquitous nucleoside adenosine deriving from ATP [40]. From a structural point of view, the A_2A_ subtype has been cloned and pharmacologically characterized, and shows seven transmembrane domains connected to three extracellular and three intracellular loops [41]. It contains a long intracellular COOH terminus presenting sites for phosphorylation and palmitoylation that may affect the process of receptor desensitization and internalization. It is present on the cell surface not only as a monomer but also in association with other receptors, e.g., A_1_ adenosine and D_2_ dopamine subtypes, forming heteromers distinguished from homomers by different functional properties [42]. The A_2A_ receptor localization affects striatum, the olfactory tubercle, and the immune system, presenting the highest expression, followed by the cerebral cortex, hippocampus, heart, lung, and vasculature. Specifically, as for neurons, astrocytes, microglia, and oligodendrocytes, they are present at pre- and postsynaptic level, regulating a series of effects associated to excitotoxicity, e.g., glutamate efflux, glial activation, and blood–brain barrier permeability, thus increasing leukocyte migration from the periphery. Concerning the peripheral immune system, A_2A_ receptors are abundant in neutrophils, monocytes, macrophages, dendritic and T cells as well in platelets, and the blood vessels, where they mediate numerous antiinflammatory, antiaggregatory, and vasodilatory effects, respectively [43]. The A_2A_ receptor signaling involves coupling to Gs and Golf proteins, in the periphery and brain, respectively, associated with adenylate cyclase and PKA stimulation leading to activation of several intracellular proteins [44,45]. In addition, it regulates MAPK signaling [46,47,48]. The ubiquitous nature of adenosine, of which the levels increase from nanomolar concentrations up to micromolar levels, following cellular and tissues damages, the wide distribution of A_2A_ receptor and their upregulation mediated by injuries, renders this subtype an interesting target for several pathologies of both the central and peripheral nervous system, including neurodegenerative and inflammatory diseases as well as cancer. 

## 4. Role of A_2A_ Adenosine Receptors in AD

### 4.1. Neuronal Injury 

Pioneering literature data report that loss of synaptic markers leading to synaptic dysfunction and degeneration is documented as one of the more important events correlated with cognitive impairment, before Aβ plaques and tangle formation [49]. More recently, it has been shown that the loss of synapses in the hippocampus and posterior cingulate gyrus is the first neuropathological alteration affecting brains of MCI and early AD patients and is an early process of memory alterations [50,51]. For this reason, AD has been defined as a synaptic-based disease, and the importance of saving synaptic structure and function has been underlined [52,53]. As for synaptic degeneration, a role for A_2A_ adenosine receptors has been recognized, recently linking it in the pathogenesis of AD [54]. Indeed, hippocampal synapses present A_2A_ adenosine receptors regulating synaptic plasticity (Figure 1) [55,56,57]. 

While A_2A_ receptor hippocampal expression is generally protective, facilitating BDNF modulation of hippocampal synaptic transmission, in aging its overexpression takes place triggering deleterious synaptic effect causing an LTP-to-LTD shift and a reduction of hippocampal-dependent learning and memory processes. This was due to an increase A_2A_-mediated glutamate release, mGluR5-dependent NMDA receptor activation, and calcium influx from overexpressed voltage-dependent calcium channels [58,59,60,61]. A similar synaptic plasticity shift, A_2A_ receptor-dependent, was observed in the hippocampus of aged and APP/PS1 animals [61,62]. It has been hypothesized that this phenomenon may be due to a shift in receptor cross-talk involving A_2A_-A_1_ heteromers, with the loss of A_2A_-mediated inhibition of presynaptic inhibitory A_1_ receptors in aged versus young rats and a direct facilitatory effect of A_2A_ stimulation [63]. Specifically, an increase in A_2A_ expression has been found in hippocampal neurons of aged or AD animal models as well as in astrocytes of AD patients and aged mice [62,64,65,66,67,68,69]. Interestingly, an increased hippocampal density of A_2A_ receptors has also been observed in AD patients [61,70,71] (Table 3). 

In addition, adenosine concentration is higher in parietal and temporal in comparison to the frontal cortex of post-mortem AD brains, thus suggesting an increased activation of A_2A_-upregulated receptors in these areas [78]. The hyperactivation of A_2A_ adenosine receptors provokes memory disabilities, LTP damage, and alterations of synaptic markers [69].

A huge body of literature demonstrates the efficacy of A_2A_ adenosine receptors antagonists, including caffeine, the world’s most popular psychoactive drug, to rescue synaptic damage and cognitive deficit in animal models of AD, proposing for them a role against synaptic toxicity [68,79,80,81,82,83]. Interestingly, chronic consumption of caffeine, or genetic deletion of A_2A_ receptors, decreases TAU hyperphosphorylation in the hippocampus, reduces neuroinflammation, and contrasts related memory deficit. Accordingly, overexpression of A_2A_ adenosine receptors increases TAU hyperphosphorylation and consequent TAU-dependent memory impairments in transgenic animal models of TAUopathy [84,85,86]. All these data are relevant because it is well known that tau pathology plays a role in memory impairment present in ageing and AD.

### 4.2. Neuroinflammation

Neuroinflammation includes a broad series of cellular effects in response to damage occurring in the nervous system as a consequence of ischemic insult, infection, and neurodegenerative pathologies. The main players of neuroinflammation are activated astrocytes and microglia that produce abnormal pro-inflammatory cytokines, such as TNF-α, IL-1β, and IFN-γ, and increase reactive oxygen and nitrogen species. It is accepted that neuroinflammation plays a crucial role in numerous neurodegenerative forms including AD, and represents an important aspect of aging that is the greatest risk factor for AD [87]. The function of cells involved in immune responses and inflammation is quite difficult to elucidate. On the one hand, microglial cells, under a moderate condition of activation, may positively destroy amyloid accumulation with beneficial effects. On the other hand, in the elderly, “inflammaging”, a chronic low-grade sterile inflammation occurring with age, induces uncontrolled production of inflammatory mediators. Indeed, this condition, characterized by a high level of cytokines, is associated with a decrease of cognition [88]. In this context, another crucial aspect of the A_2A_ adenosine receptor activation in AD concerns its regulation of neuroinflammation. Indeed, it is present in both astrocytes and microglia, regulating the secretion of pro-inflammatory cytokines, as described below (Figure 1) [89]. 

#### 4.2.1. Astrocytes

These cells include the majority of glial cells and are crucial in the regulation of brain homeostasis, due to their ability to affect synaptic plasticity, neuron metabolism, as well as ions and neurotransmitter homeostasis. Their pathological alteration is found in neurodegenerative conditions, including AD [90].

The first work documenting a role for A_2A_ receptors in astrocytes showed that they were involved in the decrease of Aβ-triggered glutamate uptake, contributing to glutamatergic synaptic dysfunction and excitotoxicity in AD [91]. This reduction was associated with a depletion of the GLT-1 glutamate transporter, Na+/K+-ATPase-dependent, that is modulated by A_2A_. Other important astrocytic functions regulated by A_2A_ receptors include calcium efflux from the endoplasmic reticulum, glutamate, and ATP release as well as GABA transport [92]. It is well known that memory is a process strictly influenced by astrocytes, where an A_2A_ receptors upregulation occurs in aging human APP mice. In this AD animal model, the conditional genetic ablation of A_2A_ receptor provides memory enhancement [68]. Accordingly, in amyloid plaque-bearing mice, administration of low doses of the A_2A_ antagonist istradefylline increased spatial memory and habituation, providing evidence that A_2A_ receptor blockers might be able to contrast memory deficits in AD patients [76]. The same effect was observed in the presence of an ENT1 inhibitor that increased adenosine and worsened memory dysfunction and neuronal plasticity in an APP/PS1 mouse model of AD [93]. Even though the pathophysiological consequence of this astrocytic A_2A_ receptor increase deserves further investigations, recent data reported that A_2A_ receptor overexpression induced important modifications, at transcriptional level, of genes involved in immune responses, angiogenesis, and cell activation [94].

#### 4.2.2. Microglia

As the brain’s resident macrophages, these cells play essential functions in the modulation of cerebral activities, by eliminating dying neurons, deleting non-functional synapses, and producing molecules important for neuronal vitality. Microglia respond to neuronal activation and prevent excessive neurostimulation, exerting a fundamental protection of the brain from excessive activation like that occurring in AD [95]. 

Evaluation of post-mortem brains revealed an overexpression of A_2A_ receptors only in microglia in proximity of pathological signs of AD but not in nondemented age-matched control brains [74]. Interestingly, it is known that neuroinflammation alone blocks neurogenesis and that contrasting inflammation reactivates this process, suggesting that the role of A_2A_ receptor in the control of neuroinflammation might be a crucial mechanism in neurodegenerative diseases [96]. As the NMDA receptor, present in both neurons and microglia, is one of the principal targets to fight AD, it is relevant that it interacts with A_2A_ receptors mainly in microglia. This interaction provides a novel functional complex, where A_2A_ blockade is useful to hamper NMDA overactivation, useful in the protection of microglial cells. Interestingly, these interacting entities were upregulated in the hippocampal cells from the APPSw, Ind mice [97]. In general, activated microglia can present two opposite phenotypes, M1 promoting inflammation and cytotoxic effects and M2 fighting inflammation and providing neuroprotection [98]. It was found that cells from APP mice presented a relevant increase of a typical M1 marker, such as inducible nitric oxide synthase (iNOS), and in M2 marker arginase-1 (Arg-1). Interestingly, the A_2A_ receptor blockade was able to decrease iNOS and increase Arg-1, providing evidence for a shift of microglia versus the beneficial M2 phenotype [97]. Accordingly, other works in animal models of neuroinflammation reported that overexpression of A_2A_ receptors caused an increase in IL-1β, IL-6, TNF-α, and typical M1 microglial markers and that its block hampered LTP deficit in hippocampus [99]. Furthermore, A_2A_ receptor constitutes heteromers with CB_2_ cannabinoid receptor subtypes, through which A_2A_ receptor antagonists may increase both endo- and exo-cannabinoid effects, providing neuroprotection [77]. 

## 5. A_2A_ Adenosine Receptor as a Novel Peripheral Biomarker in AD

Due to several pieces of evidence for the role of A_2A_ receptors to trigger synaptic and cognitive problems, it has been suggested that it might be a good candidate biomarker of some chronic neurodegenerative diseases, including AD [100,101]. Its principal localization is the striatum important for locomotor activity, followed by the frontal cortex and hippocampus, crucial for memory and cognition. Specifically, in a study evaluating A_2A_ expression in brain tissues from AD patients, an overexpression of them has been found in the hippocampus/entorhinal cortex in comparison with both frontal gray and white matter, which was not found in age-matched control brains, suggesting a relation with the presence of AD pathology [71]. Furthermore, the lowest A_2A_ presence was detected in the frontal white matter, a region less affected by AD. In other words, the pattern of expression of A_2A_ receptors seems to reflect the same distribution of AD pathology that affects the hippocampus/entorhinal cortex areas [12]. In any case, the finding of an A_2A_ overexpression in the frontal white matter, rich in glial cells, supports a role for them in glial modifications that are found in AD [71,97]. In general, it is recognized that alterations occurring in CNS pathologies may be reflected at peripheral level in the blood, representing a useful and accessible substrate to evaluate proteins playing a crucial role in the pathology [102,103]. This happens, for example, in the field of adenosine receptors, in different diseases, such as heart and respiratory failure, Parkinson, and colon cancer [104,105,106,107]. As for AD, platelets showing biochemical activities in common with neurons, such as augmented β-secretase activity as well as amyloidogenic processing of amyloid protein precursor, may be used as a peripheral model for the study of cortical pathology [108]. Interestingly, very recently, it has been demonstrated that platelets from AD patients express a higher density of A_2A_ receptors in comparison to platelets from control subjects not affected by dementia, thus providing evidence that this adenosine receptor subtype could mirror brain AD pathology in the periphery, revealing to be a promising indicator of disease [71]. These data are interesting in the diagnostic field of AD because they offer a peripheral marker that is cheap and easily accessible, through a blood withdrawal, reflecting changes in the CNS. According to the need for early diagnostic sentinels and therapeutic targets of disease, the A_2A_ adenosine receptor may satisfy both these requirements, offering a novel opportunity to find and cure AD pathology. Similarly, previous works suggested the involvement of them at the beginning of AD, with their level being higher in blood cells from patients with mild cognitive impairment (MCI) with respect to healthy subjects [109,110]. This finding should be better investigated in future studies, comparing MCI and AD patients in the same setting at different stages of the pathology to gain information about the sensitivity of the A_2A_ receptor as a biomarker of AD. In addition, other studies reported a lower amount of A_2A_ mRNA in vascular dementia versus AD patients, but no difference between these ones and control subjects [111]. All these data suggest that A_2A_ adenosine receptors may be differentially regulated in different forms of dementia, including vascular dementia and AD, probably depending on the origin/pathogenesis of the disease. 

An important issue that needs to be addressed concerns the specificity of this adenosine receptor subtype as a potential biomarker of AD. Regarding specificity in particular, an upregulation of them was already found in lymphocytes from ischemic stroke patients [112,113,114], amyotrophic lateral sclerosis and multiple sclerosis [115,116], Huntington’s and Parkinson’s disease [106,117], chronic heart failure and cardiac transplantation [105], and in PMBCs from atrial fibrillation patients [118]. In spite of this wide overexpression of A_2A_ receptors in lymphocytes and neutrophils of patients with pathologies of different nature and affecting different organs, the evidence related to their upregulation in AD has been relieved in platelets, and whether the same result is present in lymphocytes has yet to be determined. 

## 6. A_2A_ Adenosine Receptor as a Possible Therapeutic Target in AD

On the basis of the World Alzheimer Report, AD pathology is expected to rise up to more than 150 million in the next 30 years, starting from 50 million in 2018. This common type of dementia accounts for two-thirds of all dementia cases [119]. The most important risk factor for it is age, but others include both genetic and environmental factors, lifestyle, and cardiovascular pathologies. Despite many efforts employed in several years of research in disease-modifying drug development, there is no effective therapy to delay the beginning and the progression of AD. Today, the symptomatic therapies used to treat AD include nootropic molecules, reversible inhibitors of cholinesterase (donepezil or rivastigmine) improving memory and stabilizing patient’s behavior, and memantine (antagonist of NDMA receptors) producing positive effects in the cognition, behavior, and memory. Nutraceutics, containing a mix of omega-3, phospholipids, vitamins B12, B6, folate, Thiamine, vit. C and E, oligoelements, antioxidant agents (Curcumin), and coenzyme Q, are also prescribed. Unfortunately, there have been no new symptomatic drugs for 20 years, the last being memantine in 2002. A reason for this unsuccessful result may be found in the complex etiology and pathophysiology of AD.

Numerous efforts of several scientists around the world, working in the purinergic field, point to a role of A_2A_ receptor antagonists in the therapy of different neurological ailments [88,120,121,122]. Specifically, clinical studies led to the development of the new first-in-class drug istradefylline, Nourias and Nourianz, in Japan and the US, respectively, for the treatment of Parkinson’s disease [98]. Interestingly, the therapeutic value of this molecule and of the whole class of A_2A_ antagonists is not limited to improving locomotor disabilities and reducing the drawback of classical antiparkinson drugs, but also to ameliorate cognitive dysfunctions. Furthermore, their clinical development as new drugs for AD therapy may be fastened by the acquired knowledge that virtually all A_2A_ blockers, tested in animal models and clinical studies, seem very safe [83]. 

Indeed, possible drawbacks related to the wide distribution of A_2A_ receptors and their biological effects, including regulation of the immune system, inflammation, sleep, platelet aggregation and vasodilation, have not been revealed in clinical trials [123]. This is in agreement with the general safety of caffeine, by human assumption of 2–4 cup of coffee/daily, which was found able to reduce death for all causes by a recent meta-analysis [124]. Surprisingly, an exception to this safe profile of A_2A_ antagonists is represented by tozadenant, having a different chemical structure with respect to istradefylline, which induced unfortunately fatal agranulocytosis in five patients in a clinical phase III trial with 409 participants [125]. However, additional experiments are necessary to clarify if these dramatic events were due to A_2A_ blocking or other issue linked to its chemical structure.

## 7. Conclusions

Biomarkers are a primary tool in the development of AD research, but the interpretation of them should always be evaluated in light of the clinical picture. Since there are no disease-modifying therapies available on a large scale, and the diagnostic and prognostic value of biomarkers is not yet completely clear, inaccurate early diagnoses may cause deleterious psychic effects on the subjects under examination. However, beyond the early (pre-clinical) diagnosis of AD that poses such ethical problems, we believe it is possible to identify biological markers that allow us to follow the course of the disease even during its clinical phase, addressing symptomatic therapy in a personalized way. The possible use of the A_2A_ receptor as a biomarker as well as a drug target fits into this framework. Indeed, all evidence remarqued in this review underline the importance of A_2A_ receptor as a novel symptomatic drug target for AD treatment. Future clinical studies on AD patients, treated with istradefylline or with other old and new A_2A_ antagonists, will be required, and the role of A_2A_ receptor as biomarker may help to optimize patients’ enrollment and assumption protocol and dosages. In other words, molecular imaging of A_2A_ receptors in the brain or its peripheral evaluation, as a specific and sensitive platelet biomarker, may help to select patients for which the drug would be more indicated, providing the desirable approach of a personalized medicine, with a reduction of costs and wait times. Notably, this research strategy for the discovery of a new therapy for AD is of particular importance, as it may positively impact the sustainability of the system, from which both public health and the economy could take advantage.

## Figures and Tables

**Figure 1 cells-10-02344-f001:**
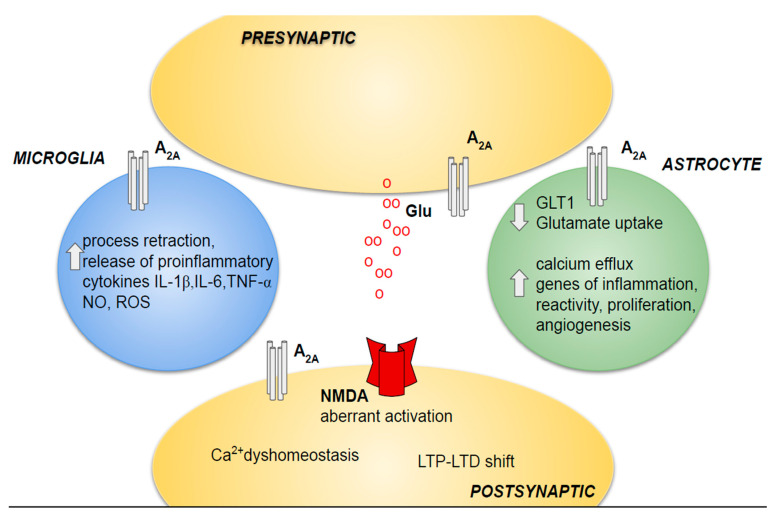
Schematic diagram illustrating the role of A_2A_ adenosine receptors in AD.

**Table 1 cells-10-02344-t001:** Individual score obtained from the combination of ATN [15,16].

ATN System	
A-T-N-	No biomarkers of degenerative brain pathology
A + T-N-	Amyloid deposition (Alzheimer’s continuum)
A + T-N+	Amyloid deposition and non-tau degeneration (Alzheimer’s continuum; suspected non-AD pathology)
A + T + N-	Early Alzheimer’s Disease (Alzheimer’s continuum)
A + T + N+	Alzheimer’s Disease (Alzheimer’s continuum)
A-T + N- A-T-N+ A-T + N+	Non-Alzheimer neurodegenerative Diseases (e.g., primary TAUpahies, Fronto-Temporal Dementia due to TDP-43, LATE, other rare forms)

**Table 2 cells-10-02344-t002:** Biomarkers of Alzheimer’s Disease.

Biomarkers of Alzheimer’s Disease	
ATN System [15,16]	
Biomarkers of β-amyloid plaques (A)	Cortical amyloid PET
	Low CSF β-amyloid 42
Biomarkers of tau (T)	Cortical tau PET
	Elevated CSF phospho-tau
Biomarkers of neurodegeneration and neuronal injury (N)	[(18)F]-fluorodeoxyglucose PET hypometabolism
	Atrophy on MRI
	Elevated CSF total-tau
**Fluid biomarkers**	
Increased levels of plasma Neurofilament light (NfL) [22,27]	
Increased levels of plasma tau phosphorylated at threonine 181 (P-tau181) [23]	
Increased levels of plasma tau phosphorylated at threonine-217 (P-tau217) [28]	
Increased levels of plasma Aβ-amyloid oligomers [24]	
Lower levels of plasma Aβ42/Aβ40 ratio (TP42/40) [25,26]	
Increased levels of CSF Neurogranin (Ng) [29]	
Increased levels of CSF YKL-40 [34,35]	

**Table 3 cells-10-02344-t003:** Upregulation of A_2A_ adenosine receptors in aged or animal models of AD and in AD patients.

Aged/Animal Models of AD/AD Patients	Tissue/Cell	References
aged rats	cortex and hippocampus	[72]
aged vs. young rats	cortical membranes	[64]
aged vs. young rats	hippocampal neurons	[57,73]
AD patients	microglia	[74]
APPsw tg mice	hyppocampus	[65]
adult and aged rats	hyppocampus	[75]
AD patients	frontal cortex	[70]
Rats with different ages	nerve terminals purified from the hippocampus	[66]
AD rat model	hippocampus	[67]
AD patients/aged mice	astrocytes	[68,76]
APP/PS1 mice	CA3 synaptic membranes	[62]
Early AD mice model	hippocampal synaptosomes	[69]
APP_Sw/Ind_ AD transgenic mice model	microglia	[77]
Aged subjects and AD patients	hippocampal neurons	[60]
AD patients	cortex, hippocampus, platelets	[71]

Abbreviations: swedish mutation (Sw) transgenic (Tg), Indiana (Ind) mutations.

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
