# Peer review of "A2A Adenosine Receptor as a Potential Biomarker and a Possible Therapeutic Target in Alzheimer’s Disease"

_cells, 2021, doi:10.3390/cells10092344_

Round 1

Reviewer 1 Report

I believe that the revised form of the manuscript is suitable for publication.

Reviewer 2 Report

This is very interesting work, discussing timly problem related with neuordegenerative diseases - lack of fast and not invasive for testing biomarkers. Authors are giving convincing reasons that A2A adenosine receptors located in periphery may be a good biomarker of Alzheimer disease. 

Manuscript is very well written, I only found small errors i Table 3: "hyppocampus" need to be corrected.

Reviewer 3 Report

Dear authors,

the review accomplished with the expectations, being exhaustive and well written. 

Just one suggestion, separate the table into two tables. Human and animal models.

Congratulations.

This manuscript is a resubmission of an earlier submission. The following is a list of the peer review reports and author responses from that submission.

Round 1

Reviewer 1 Report

I examined the work of Gessi et al, entitled 'A2A Adenosine Receptor as a Potential Biomarkerand a Possible Therapeutic Target in Alzheimer's Disease' and I consider it valuable and of potential interest for many professionals in the AD field.

However, I think there are a few issues that the authors should address before their work might be published.

Major issues:

1) A more detailed discussion about sensitivity and specificity about AD biomarkers, including A2A adenosine receptor, is needed.

2) A figure illustrating the role of A2A adenosine receptors in AD is needed.

Minor issues:

3) page 3 - ATN system: it might not be clear for the reader how A-T-N+ is exemplified as primary tauopathies, FTP, etc. - maybe an example of neurodegeneration not involving tau is worthy to be mentioned

4)it would be interesting for the discussion to develop a discussion about how biomarkers molecules are released and cleared in the tissues/fluids.

5) I would be interested to see a statement about possible side effects of antagonizing the A2A receptors.

Reviewer 2 Report

The authors summarize the current knowledge on the biomarkers for AD and propose the A2A AR as another, potential biomarker for the disease. They also discuss possibility of using A2A as a biological target for new drugs against  AD. I advise publication of this work (finding therapies to control AD would be a blessing for humanity) after following amendments:

In order to maintain the thematic consistency of the article and to avoid repetition of information already published the chapter 1 “Alzheimer’s disease” should be shortened as some fragments/phrases has been published elsewhere (e.g. Front. Pharmacol., 22 March 2021 | https://doi.org/10.3389/fphar.2021.652455)

There are three chapters no. 1 in the work. Numbering should be changed.

The biology of adenosine A2A receptors in the context of AD should be adequately described in "A2A adenosine receptors biology". It would be nice to see a schematic diagram illustrating the targeting and localization of the A2A receptor in neurons, astrocytes and microglia.

ARs are considered attractive biological targets for new drugs; however, few drugs based on this concept are in clinical practice.There is no section on pharmacologically active compounds targeting A2A as candidates for anti-AD therapy in the manuscript. This chapter would be useful. Without it, the work seem to be too general and not very concise.